# The Yeast Gsk-3 Kinase Mck1 Is Necessary for Cell Wall Remodeling in Glucose-Starved and Cell Wall-Stressed Cells

**DOI:** 10.3390/ijms26083534

**Published:** 2025-04-09

**Authors:** Fan Zhang, Yingzhi Tang, Houjiang Zhou, Kaiqiang Li, James A. West, Julian L. Griffin, Kathryn S. Lilley, Nianshu Zhang

**Affiliations:** 1Department of Biochemistry, University of Cambridge, 80 Tennis Court Road, Cambridge CB2 1GA, UK; isabellaz2@sjtu.edu.cn (F.Z.); kl470@cantab.ac.uk (K.L.);; 2State Key Laboratory of Microbial Metabolism, Joint International Research Laboratory of Metabolic and Developmental Sciences, School of Life Sciences and Biotechnology, Shanghai Jiao Tong University, Shanghai 200240, China; 3MRC Protein Phosphorylation and Ubiquitylation Unit, Sir James Black Centre, School of Life Sciences, University of Dundee, Dundee DD1 5EH, UK

**Keywords:** Mck1, Slt2, SNF1, SAGA, PKA, polarized growth, metabolic reprogramming

## Abstract

The cell wall integrity (CWI) pathway is responsible for transcriptional regulation of cell wall remodeling in response to cell wall stress. How cell wall remodeling mediated by the CWI pathway is effected by inputs from other signaling pathways is not well understood. Here, we demonstrate that the Mck1 kinase cooperates with Slt2, the MAP kinase of the CWI pathway, to promote cell wall thickening in glucose-starved cells. Integrative analyses of the transcriptome, proteome and metabolic profiling indicate that Mck1 is required for the accumulation of UDP-glucose (UDPG), the substrate for β-glucan synthesis, through the activation of two regulons: the Msn2/4-dependent stress response and the Cat8-/Adr1-mediated metabolic reprogram dependent on the SNF1 complex. Analysis of the phosphoproteome suggests that similar to mammalian Gsk-3 kinases, Mck1 is involved in the regulation of cytoskeleton-dependent cellular processes, metabolism, signaling and transcription. Specifically, Mck1 may be implicated in the Snf1-dependent metabolic reprogram through PKA inhibition and SAGA (Spt-Ada-Gcn5 acetyltransferase)-mediated transcription activation, a hypothesis further underscored by the significant overlap between the Mck1- and Gcn5-activated transcriptomes. Phenotypic analysis also supports the roles of Mck1 in actin cytoskeleton-mediated exocytosis to ensure plasma membrane homeostasis and cell wall remodeling in cell wall-stressed cells. Together, these findings not only reveal the novel functions of Mck1 in metabolic reprogramming and polarized growth but also provide valuable omics resources for future studies to uncover the underlying mechanisms of Mck1 and other Gsk-3 kinases in cell growth and stress response.

## 1. Introduction

The fungal cell wall acts as a primary physical barrier to the external environment and is essential for cell viability, morphogenesis and pathogenesis [1]. In the yeast *S. cerevisiae*, the cell wall represents up to 30% of a cell’s dry weight and comprises two layers that are distinguishable by ultrathin-sectioning electron microscopy: an electron-transparent inner layer of crosslinked β-1,3-glucan, β-1,6-glucan and chitin and an electron-dense outer layer of mannoproteins [2,3,4]. Most of the mechanical strength of the cell wall is derived from its inner layer, in which the β-linked glucans are the major components (>95%) of the polysaccharide fraction [2]. The cell wall integrity (CWI) pathway, together with the HOG (high-osmolarity glycerol) and the calcium–calcineurin pathways, is responsible for transcriptional regulation of cell wall synthesis during growth and cell wall remodeling in response to cell wall stress conditions [5,6]. Cell wall stress is sensed by three classes of plasma membrane proteins (Wsc1-3, Mid2 and Mtl1), which interact with the guanine nucleotide exchange factor (GEF) Rom1/2 activating the small GTPase Rho1. Rho1 activates protein kinase C 1 (Pkc1), which subsequently triggers the conserved MAPK module composed of Bck1, Mkk1/Mkk2 and Slt2. The MAP kinase Slt2 (also known as Mpk1) then launches the transcriptional response through the transcription factor Rlm1 and the chromatin modifiers SWI/SNF and SAGA [5,7,8].

Transcription activation mediated by the CWI pathway in response to cell wall damage is a key part of the ‘compensatory salvage response’ program involving both synthesis and cross-linking of the cell wall polymers [3,6,9]. In contrast, cell wall thickening in the post-diauxic shift (PDS) cells, resulting partially through the enhanced expression of β-1,3-glucan synthase Gsc2 and localized synthesis of UDPG [4], does not involve many changes in the crosslinking among the polysaccharides [10,11]. A number of nutrient and energy sensing and signaling pathways, including the TOR, PKA and the SNF1 complex, have been implicated in coordinating cell wall thickening and the acquisition of other characteristics of stationary-phase cells, including enhanced stress resistance and the accumulation of storage carbohydrates [12,13,14]. Snf1 (the AMP-regulated kinase in the SNF1 complex) is essential to respiratory cell growth in part through the activation of the metabolic reprogramming mediated by the Cat8 and Adr1 transcription activators [15,16,17,18,19] and is also an important regulator of UDPG partitioning between the structural and storage carbohydrates in PDS cells [20,21,22]. While it is known that UDPG levels are increased in cells starved of glucose [23], the involvement of signaling pathways other than Snf1 in UDPG synthesis in PDS cells remains unclear.

We have previously demonstrated that among the four Gsk-3 homologues in yeast [24], Mck1 is a key regulator of stationary phase entry and exit [25]. In response to glucose starvation, Mck1 cooperates with the two TORC1-/PKA-negatively regulated kinases, the Greatwall family kinase Rim15 [26,27] and the DYRK kinase Yak1 [28] to promote G_0_ entry, the acquisition of stress resistance, the accumulation of storage carbohydrates and the extension of chronological lifespan [25,29,30,31]. Interestingly, the stationary-phase *mck1Δrim15Δ* double mutants demonstrated a much lower cell viability than either *mck1Δ* or *rim15Δ* single knockouts, as determined by Sytox green staining of the nuclear DNA [29], suggesting that the *mck1Δrim15Δ* cells may also bear structural defects entering the stationary phase. In this study, we initially confirmed that Mck1 and, to a lesser degree, Rim15 act in parallel to Slt2 to promote cell wall thickening in PDS cells. Focusing on Mck1 and using a multi-omics approach, we then revealed that Mck1 is required for UDPG accumulation and cell wall thickening in PDS cells through the activation of the Msn2/4-dependent stress response and the Snf1-directed metabolic reprogram. Comprehensive analysis of the phosphoproteome suggested that Mck1 may be involved in the Snf1-directed metabolic reprogram through PKA inhibition and SAGA-mediated transcription activation, while Mck1 is necessary for plasma membrane/cell wall remodeling in cell wall-stressed cells through its roles in actin cytoskeleton-mediated exocytosis. Gene ontology (GO) analysis indicated that Mck1-mediated phosphoproteins fall into similar functional categories as mammalian Gsk-3 substrates. These findings not only shed novel insights into the cell wall remodeling mechanisms under different stress conditions in yeast but also provide valuable resources for future studies to reveal the pleiotropic effects and the underlying mechanisms of Mck1 and other Gsk3 kinases in cell growth and stress response.

## 2. Results

### 2.1. Mck1 Cooperates with Rim15 and Slt2 to Regulate Cell Wall Thickening in PDS Cells

The structural defects exhibited by the stationary-phase *mck1Δrim15Δ* double mutants [29] suggest that Mck1 and Rim15 may cooperate to regulate cell wall integrity during the transition phases. To reveal whether Mck1 cooperates with Rim15 and the CWI pathway to regulate cell wall thickening in PDS cells, WT, single, double and triple knockouts of *MCK1*, *RIM15* and *SLT2* were grown in YPD (2% glucose) to the early stationary phase (Figure 1A). Glucose was consumed between 11 and 12 h in all samples (diauxie, 12 h). The cell wall thickness (the β-glucan layer) was measured by Transmission Electron Microscopy (TEM) imaging at both the mid-exponential phase (EXP) and the early-stationary phase (SP, Figure 1A). TEM images of at least fifty cells were taken to quantify cell wall thickness at both growth phases (Figure 1B and Appendix A). As expected, WT cells increased their cell wall thicknesses from EXP to SP by ~2.5 fold (Figure 1B,C). The increase in cell wall thickness was compromised in all of the three single mutants, with the cell wall of the *slt2Δ* or *rim15Δ* cells thickened by ~2 fold and that of the *mck1Δ* cells by ~1.6 fold (Figure 1C). The *mck1Δrim15Δ* double mutants displayed a thinner cell wall than either single mutants at SP (Figure 1C), confirming that Mck1 and Rim15 cooperate to promote cell wall thickening in PDS cells.

Among the three double mutants, the thickness of the *mck1Δslt2Δ* cell wall was the least increased (Figure 1C), indicating that Mck1 and Slt2 may act in parallel to promote cell wall thickening during the transition into SP. Interestingly, a slight increase in cell wall thickness was observed in the EXP *mck1Δ* cells (Figure 1C). Such increase was not seen in the EXP *mck1Δrim15Δslt2Δ* triple mutants (Figure 1C), suggesting that deletion of *MCK1* may lead to isotropic cell wall synthesis in a manner dependent on Rim15 and Slt2 (see later sections). Nevertheless, removal of *MCK1* further compromised the cell wall thickness of the *rim15Δslt2Δ* double mutants grown to SP (Figure 1C), supporting the essential roles of Mck1 in cell wall thickening in PDS cells. We next focused on Mck1 to further characterize its roles and its genetic relationship with Slt2 in cell wall thickening in PDS cells.

### 2.2. Mck1 Is Responsible for UDPG Accumulation in PDS Cells

The synthesis of trehalose, glycogen and cell wall β-glucan shares a common substrate of UDPG. The accumulation of storage carbohydrates is compromised in the *mck1Δ* cells [25]. Thus, we hypothesized that Mck1 may be responsible for the synthesis of UDPG or its precursors in PDS cells to accumulate storage and structural carbohydrates. To confirm this hypothesis, the levels of UDPG in WT, single and double mutants of *MCK1* and *SLT2* were quantified in EXP, diauxie and mid-PDS cells (Figure 1A). Sugar nucleotides were extracted and analyzed by liquid chromatography tandem mass spectrometry (LC-MS^2^). As shown previously [23], the levels of UDPG were increased in WT cells grown to mid-PDS (Figure 2A). The relative levels of UDPG were not significantly regulated by *MCK1* and/or *SLT2* in EXP cells (Figure 2B). Relative UDPG levels were decreased dramatically in the *mck1Δ* mutants but not significantly changed in the *slt2Δ* cells upon diauxie (Figure 2C). Very low levels of UDPG were similarly detected in the *mck1Δslt2Δ* mutants as observed in the *mck1Δ* single mutants, indicating that UDPG accumulation is dominantly regulated by Mck1 in glucose-depleted cells. Compared to diauxie, relative UDPG levels were increased in the mid-PDS *mck1Δ* cells but still significantly lower than that in WT cells (Figure 2D). Using only the mid-PDS cells and a recently improved method to extract the TCA cycle metabolites (see Section 4), the levels of G1P (glucose-1-phosphate) and G6P (glucose-6-phosphate) together and UDPG were shown to be decreased in the *mck1Δ* cells (Figure 2E). The levels of the TCA cycle intermediates in the *mck1Δ* mutants, especially fumarate and malate, were also significantly lower than those detected in WT cells (Figure 2F), suggesting that Mck1 may be required to promote mitochondrial metabolism and gluconeogenesis, leading to UDPG accumulation and cell wall thickening (Figure 1C).

### 2.3. MCK1 Is Necessary to Promote Gene Expression Implicated in the Stress Response, TCA Cycle and Gluconeogenesis

To reveal how Mck1 may promote UDPG accumulation in PDS cells, RNAseq analyses of the transcriptome in WT, *mck1Δ*, *slt2Δ*, and *mck1Δslt2Δ* cells grown at EXP and early-PDS (diauxie + 1h, Figure 1A) were conducted. Principal component analysis indicated that all the EXP samples, regardless of their genotypes, were closely clustered (encircled in Figure 3A), suggesting that *MCK1* or *SLT2* does not have a significant impact on the transcriptome of the actively growing cells. The early-PDS (labeled as PDS hereafter) samples, however, were clearly separated by *mck1Δ* rather than *slt2Δ* (Figure 3A), indicating that Mck1 regulates a glucose depletion-induced gene expression program on which Slt2 has a minimal impact. The expression of 608 genes was found to be significantly regulated by *MCK1* in PDS cells (FDR < 0.05, ≥1.5 fold). K-means hierarchical clustering of these differentially expressed (DE) genes (listed in Appendix A) identified four major clusters (Figure 3B). Relative expression levels of these DE genes did not seem to be influenced by *slt2Δ* (Figure 3B), further supporting that Mck1 may mediate a gene expression program largely independent of Slt2 (Figure 3A).

Among the genes repressed in WT PDS cells (clusters 1 and 2, Figure 3B), *MCK1* was necessary to maintain their moderate expression levels (cluster 1) or to ensure that their expression levels were kept low (cluster 2). Similarly, *MCK1* was required to promote the expression of those strongly activated by glucose depletion (cluster 3) or to prevent their expression levels from hyperactivation (cluster 4). Enriched GO terms (process), motifs and potential transcription factors (TFs) were identified for each cluster (Figure 3B). Genes implicated in ribosome biogenesis (RiBi) were found to be overrepresented in cluster 1, and the motifs targeted by TFs Sfp1 and Stb3 were enriched in their promoter regions by the RSAT analysis [32]. Stb3 is a ribosomal RNA processing element (AAAWTTTT)-binding protein responsible for the repression of RiBi and cell growth in glucose-starved cells through recruiting the histone deacetylase complex RPD3L [33,34,35,36]. Sfp1, responsible for transcription activation of RiBi and other growth-related genes in glucose-replete cells [37,38], was shown to be required for the expression of most of the cluster 1 genes by Yeastract+ analysis [39]. These data suggest that Mck1 may function to restrict the repression activity of Stb3 and/or promote the activation activity of Stp1 to maintain the translation machinery at a certain level in PDS cells (cluster 1, Figure 3B).

Genes involved in different metabolic processes were enriched in all other clusters (2, 3 and 4, Figure 3B), implicating Mck1 in promoting carboxylic acid metabolism (cluster 3) and concurrently restricting carbohydrate (cluster 2) and disaccharide (cluster 4) metabolic processes. A number of TFs were also identified to be responsible for the transcription of the cluster 3 genes (Figure 3B). These include the general stress response factors Msn2/4, the post-diauxic shift factor Gis1, and the carbon source responsive transcription activators Adr1 and Cat8. We have previously demonstrated that glucose starvation-induced expression of heat shock proteins, such as *HSP26* and *SSA3*, is dependent on Mck1 [29]. Adr1 and Cat8, activated by the SNF1 complex, have been shown to promote the transcription of genes required for the utilization of non-fermentable carbon sources [19]. Indeed, pathway enrichment analysis of the DE genes indicated that those involved in the TCA cycle (*PYC2*, *ACO1*/*2*, *IDH1*, *IDP1*, *LSC2*, *SHH4*, *FUM1*), gluconeogenesis (*FBP1* and *PCK1*), the pentose phosphate pathway (*GND2*, *TKL2*, *NQM1* and *PRS3*) and *CAT8* itself were down-regulated in the *mck1Δ* cells (cluster 3, Appendix A). Conversely, the glucose transporter genes (*HXT3* and *HXT4*) and the unidirectional glycolytic genes, *HXK2*, *PFK27* and *CDC19*, were seen to be up-regulated in the *mck1Δ* mutants (cluster 2, Appendix A). These data lend further support to the idea that *MCK1* is necessary to activate the stress response regulon and the Cat8- and Adr1-dependent metabolic program and, concomitantly, for attenuating the glycolysis pathway in response to glucose depletion.

To further confirm the above observations, endogenous *FBP1* and *PCK1* were fused with GFP at their genomic loci. Glucose depletion triggered a significant increase in Fbp1-GFP (~2.5-fold) and Pck1-GFP (~2-fold) in WT cells (Figure 3C). The increase in both GFP reporters was abolished in the *mck1Δ* mutants, confirming that the gluconeogenesis pathway is positively regulated by Mck1. Intriguingly, compared to WT, the level of Fbp1-GFP was seen to be significantly increased in the exponentially growing *mck1Δ* mutants (5.5 h, Figure 3C), suggesting the roles of Mck1 in restricting the basal levels of Fbp1 expression. Nevertheless, these reporter assays confirmed the roles of Mck1 in facilitating the Cat8- and Adr1-dependent metabolic program in response to glucose depletion.

### 2.4. Integrative Analysis of the Proteome and the Transcriptome Confirms the Transcriptional Control of the Gluconeogenesis and Stress Response by Mck1

To reveal how Mck1 may be implicated in glucose depletion-induced metabolic reprogramming, we also determined the genes that are regulated by Mck1 at the protein level using a TMT 11-plex labeling kit. Total proteins were isolated from the same samples used for transcriptome studies. Compared to the transcriptome (excluding the Slt2 data from analysis, Figure 4A), PCA indicated that Mck1 is implicated in the regulation of the proteome in both EXP and PDS cells (Figure 4B), suggesting post-transcriptional regulation of gene expression by Mck1. In total, 612 transcripts and 190 proteins were shown to be differentially regulated by Mck1 by more than 1.5-fold (FDR < 0.05) in PDS cells. Removing genes missing transcript (6) or protein (193) data left 558 of them that are significantly regulated by Mck1 at the transcript and/or protein levels (Appendix A). Poor correlation (Pearson correlation coefficient: 0.37) was observed between the transcript and protein regulation levels (Figure 4C), further suggesting that Mck1 may be involved in gene expression at both the transcriptional and post-transcriptional levels. Nevertheless, genes implicated in the stress response, mitochondrial function, glyoxylate cycle and gluconeogenesis were down-regulated in the *mck1Δ* mutants at both the transcript and protein levels (Table 1 and the red symbols on the bottom left of Figure 4C). Among the 53 genes that were shown to be bound and transcriptionally activated by the Cat8 and/or Adr1 transcription factors [40], 15 of them were found in Table 1 (labeled with *). The protein levels of *CAT8* itself were also seen to be significantly decreased in the *mck1Δ* mutants (Table 1). These data support the claim that the Cat8- and Adr1-mediated metabolic reprogram and the Msn2/4- and Gis1-dependent stress response regulon are activated by Mck1 at least in part through transcription regulation. In addition, both the transcript and protein levels of the hexose transporter *HXT1*, *HXT3* and *HXT4* and the predominant hexose kinase *HXK2* were enhanced in the *mck1Δ* mutants (red symbols on the top right of Figure 4C). Besides its roles in glycolysis, Hxk2 has been proposed to form a complex with the transcription repressor Mig1 in glucose-replete cells to repress gene expression required for growth on alternate carbon sources [41,42]. However, a recent publication disputes the model in which Hxk2 is localized to the nucleus in glucose-replete cells or acts as a transcription repressor of the Snf1-directed gene expression program [43]. Nevertheless, the activation of the Snf1-dependent metabolic program and concurrent down-regulation of the glycolysis pathway (Figure 3B and Figure 4C) suggest that Mck1 may target the signaling pathways or their downstream effectors responsible for the regulation of fermentative and respiratory growth.

### 2.5. Analysis of the Mck1-Mediated Phosphoproteome Reveals the Conserved Sequence Motifs Targeted by the Gsk-3 Kinases

To identify the potential targets of Mck1 that are accountable for the metabolic reprogram, we conducted phosphoproteome studies using the same proteome samples (EXP and PDS (early-PDS)). More than thirty-three thousand phosphopeptides were quantified across the 11 samples. PCAs of the phosphopoteome at the phosphorylation (before normalization to the proteome) and the occupancy (after normalization to the proteome) levels revealed similar patterns of separation between WT and *mck1Δ* mutants at both growth phases (Appendix A). In the *mck1Δ* mutants, the phosphorylation levels of 815 and 336 peptides were, respectively, reduced and increased by more than 2-fold (Figure 5A). In total, 83% of the 815 peptides were among the 771 phosphopeptides similarly decreased at the occupancy levels (Figure 5B,C). Gsk3-mediated phosphorylation has been linked to protein instability in mammals [44]. Among the 91 proteins whose abundance was enhanced in the *mck1Δ* mutants (Figure 4C), Mck1-mediated phosphorylation (Appendix A) was only revealed in two of them (Hlr1 and Mkt1, circled in Figure 4C). These data indicated that the vast majority of the Mck1-mediated phosphorylations may not lead to protein instability, at least in the population of cells grown to glucose depletion. Therefore, we decided to use the phosphoproteome without normalization to the proteome to further characterize the potential phosphotargets of Mck1 (Figure 5A).

GSK-3 substrates generally bear a consensus sequence (S/T)XXX(S/T) and tend to be phosphorylated by other “priming” protein kinases at the C-terminal (+4) S/T prior to being phosphorylated by GSK-3 at the first S/T [45,46]. The 815 peptides with significantly reduced phosphorylation levels correspond to 271 proteins (Appendix A). Analysis of the peptide sequence using the MEME suite [47] revealed a number of enriched motifs, all but one matching the afore-mentioned consensus sequence (Figure 5D). pS/pT followed by Pro was found in two motifs, suggesting that phosphorylation by one or more proline-directed kinases in the MAPK and CDK families [48,49] may be compromised in the *mck1Δ* mutants. At the protein level, phosphorylation at the first S/T within the consensus sequence was found at least once in 116 of the 271 proteins (Appendix A), including a number of previously confirmed phosphotargets of Mck1, such as Rcn1^S113 S117^ [50], Rpc53^S224 T228 T232^ [51], Elo2^S1325 S1328 S1329^ [52] and Hsl1 [53]. Mck1 and calcineurin have been shown to coordinately regulate the destabilization of Hsl1 to delay the onset of mitosis in yeast cells exposed to high concentrations of Ca^2+^. The phosphosites in Hsl1, not revealed previously, were identified at three residues, S^1325^, S^1328^ and S^1329^, in our study (Appendix A). Furthermore, among the 73 Mck1-dependnent phosphoproteins revealed by Bodenmiller et al. [54], 17 of them bearing one or more identical phosphosites were also found in the list of the 271 phosphoproteins. The enrichment of the Gsk-3 target motif in the phosphopeptides and the identification of the previously confirmed targets led us to conclude that we have revealed a comprehensive list of Mck1 phosphotargets in vivo (Appendix A).

Interestingly, analysis of the 336 peptides (corresponding to 174 proteins) whose phosphorylation levels were significantly enhanced in the *mck1Δ* mutants (Figure 5A) revealed the same enriched motifs, with the phosphosites being the C-terminal S/T (Figure 5E). Out of the 116 motif-bearing proteins with reduced phosphorylation at the first S/T, enhanced phosphorylation at the C-terminal S/T, mostly monophosphorylation (multiplicity = 1), was identified in 42 of them (Appendix A). Furthermore, more than 70% of the phosphorylations at the first S/T were found only in dual- and/or multi-phosphopeptides (multiplicity ≥ 2) (Appendix A). These data strongly suggest that like other Gsk-3 kinases, Mck1 may also prefer substrates which have been prephosphorylated at the +4 S/T by either the proline-directed or -3 Arg-selecting kinases (Figure 5E).

### 2.6. Mck1 May Be Involved in the Snf1-Directed Metabolic Reprogram Through the SAGA Coactivator and PKA Inhibition

Mck1 has been shown to activate the Msn2/4-dependent gene expression through promoting their nuclear localization under environmental stress conditions [55,56]. To find how Mck1 may be involved in the activation of the Snf1-mediated metabolic reprogram, we initially conducted GO analysis of the 271 Mck1-dependent phosphotargets (Appendix A). Genes in the category of “sites of polarized growth” (by component) were significantly enriched (FDR: 1.04 × 10^−18^). Similar GO analysis (by process) also revealed the enrichment of genes in signal transduction (FDR: 9.9 × 10^−12^), mitotic cell cycle (FDR: 9.3 × 10^−12^) and Pol II gene transcription (FDR: 2.1 × 10^−8^). Those bearing the consensus motif (S/TXXXS/T) in the four categories are listed in Table 2, including the phosphosites exhibiting enhanced phosphorylation levels at the C-terminal S/T.

The Mck1-mediated phosphoproteins implicated in signaling and Pol II transcription included two (Sip1 and Sip2) of the three alternate β-subunits of the heterotrimeric SNF1 complex, three subunits (Sgf29, Spt20 and Taf5) of the SAGA complex and two Not proteins (Not3 and Not5) in the Ccr4-Not complex (Table 2). The β-subunits of the SNF1 complex confer functional specificity to the Snf1 kinase [57,58], and the Sip2-Snf1 kinase is responsible for the major fraction of Snf1 activity during growth on nonfermentable carbon sources [59]. In the SAGA complex, Sgf29 is a component of the HAT module and Spt20 is a subunit of the transcriptional regulatory complex of SAGA, while Taf5 is an essential protein involved in chromatin organization, histone acetylation and Pol II transcription [60,61]. SAGA recruitment and H3 acetylation are crucial to transcription activation of the Cat8- and Adr1-mediated genes, including *ADH2*, *FBP1* and *PCK1*, in glucose-depleting cells [62,63,64,65]. SAGA also functions as a scaffold to recruit other coactivators and in return impacts the stable binding of Cat8 [66,67]. Among the 182 genes dependent on Mck1 for transcription activation in glucose-depleted cells (cluster 3, Figure 3B), transcription up-regulation of 79 of them, including those mediated by the Cat8 and Adr1 transcription activators (Figure 6A), was shown to rely on Gcn5, the catalytic subunit of SAGA [62]. Furthermore, the highly conserved Ccr4–Not complex is necessary for SAGA-dependent H3 acetylation and gene expression [68,69,70,71]. These studies and our data imply that Mck1-mediated phosphorylation of the multiple subunits of the SNF1, SAGA and Ccr4-Not complexes (Table 2) may act together to promote transcription activation of the Snf1-directed metabolic reprogram in which SAGA functions as a coactivator.

To find additional targets of Mck1 responsible for the activation of the metabolic reprogram, we also conducted motif and GO analyses of the 785 phosphopeptides (corresponding to 286 proteins) down-regulated in the *mck1Δ* mutants only in glucose-depleted cells (Appendix A). Surprisingly, none of the enriched motifs bear the consensus sequence with the phosphosite at the first S/T (Figure 6B). Instead, all the overrepresented motifs are either pS/pT followed by Pro or bear an Arg at the -3 position (Figure 6B), suggesting that the activities of some proline-directed and -3 Arg-selecting kinases may be down-regulated in the glucose-depleted *mck1Δ* cells. GO analysis (by process) of the 286 phosphoproteins revealed the enrichment of genes in endocytosis (FDR: 2.4 × 10^−6^), intracellular signaling cassette (FDR: 2.1 × 10^−5^), actin filament-based process (FDR: 6.8 × 10^−5^) and others. Strikingly, 5 out of the 17 proteins in the category of “intracellular signaling cassette” are involved in the PKA signaling pathway (Figure 6C), including Cdc25 and Ira1 which are, respectively, a GEF (GTP/GDP exchange factor) and a GAP (GTPase activating protein) for Ras proteins, the adenylate cyclase Cyr1 required for cAMP production, and the motif-bearing Tpk1 and Bcy1, functioning, respectively, as the catalytic and regulatory subunits of the tetrameric PKA complex. Hyperphosphorylated Bcy1 mediated in part by Mck1 has been shown to be retained in the cytoplasm in heat-stressed cells [72]. Phosphorylation of Bcy1 (S^70^) by Mck1 (Figure 6C) has also been demonstrated by Bodenmiller et al. [54]. Furthermore, the catalytic active Mck1 has been shown to inhibit the activity of Tpk1 in vitro in the presence or absence of Bcy1 [73]. These studies and our data suggest that Mck1 may be involved in phosphorylating multiple components of the PKA pathway to negatively regulate PKA activity in glucose-depleted cells.

Previous studies have demonstrated that the function of the SNF1 complex is negatively regulated by PKA in a number of ways. PKA phosphorylates Sak1, one of the three Snf1-activating kinases, and contributes to the regulation of Snf1 activity [74]. Sip1 phosphorylation by PKA prevents the localization of the SNF1 complex to the vacuole [75]. Furthermore, PKA phosphorylates and deactivates the transcriptional activator Adr1, whereas Snf1 indirectly causes its dephosphorylation and activation [76,77]. Thus, deletion of *MCK1* may lead to improper PKA activation, resulting in lower SNF1 activity and hence compromised gene expression dependent on Snf1/SAGA (Figure 6A). Improper PKA activation may also lead to decreased activities of the Rim15 and Yak1 kinases, which have been shown to be negatively controlled by PKA (see Section 1). Snf1, Rim15 and Yak1 have all been classified as -3 Arg-selecting kinases [48], corresponding to the -3 Arg motifs enriched in the Mck1-mediated phosphopeptides revealed specifically in glucose-depleted cells (Figure 6B).

### 2.7. Mck1 Phosphotargets Are Implicated in Polarized Growth and Cytokinesis

The above GO analyses of the Mck1-mediated phosphoproteins in both growth phases (Appendix A) and those specifically revealed in PDS cells (Appendix A) both suggested the roles of Mck1 in actin cytoskeleton-mediated polarized growth. We focused on the motif-bearing phosphoproteins regulated by Mck1 in both growth phases (Table 2). Eleven of the twenty-eight phosphoproteins involved in polarized growth were also found in the category of “mitotic cell cycle” (Table 2), suggesting that Mck1 may be involved in coordinating polarized growth with the progression of the cell cycle. The highly conserved molecular machinery centered on the small GTPase Cdc42 regulates cell polarity in diverse organisms [78,79,80]. In *S. cerevisiae*, Cdc42 is activated by upstream GEFs and initiates polarized growth through a number of downstream effectors, including formins, PAKs (p21-activated kinases), scaffold proteins and the exocyst [81,82,83]. Indeed, the Mck1 phosphotargets include Bud3, a GEF activating Cdc42 to establish a budding site [84]; Ste20, a PAK involved in bud site selection and polarized growth [85,86,87]; and a number of effectors involved in polarized exocytosis (Figure 7A). Bnr1 and Bni1 are formins that are responsible for nucleating the assembly of actin cables for vesicle and mRNA transport [88,89]. Smy1 is a kinesin-related protein known to promote the association of myosin-V with secretory cargo [90]. Boi1 and Boi2 are necessary for directing an actin nucleation complex to sites of exocytosis [91,92]. Furthermore, a number of Mck1 phosphotargets have been implicated in actin cytoskeleton-mediated endocytosis (Figure 7B and Table 2). The Pkh2 kinase in the sphingolipid-mediated signaling pathway, together with Pkh1, controls eisosome assembly and turnover [93,94]. The Ark1/Prk1 family kinases Prk1 and Akl1, the latter of which transmits TORC2 signals, control endocytic machinery, together with Ark1, the other kinase in the family [95,96,97]. Sla1, a substrate of Akl1, is an adaptor protein which interacts with clathrin to control coat formation, cytoskeleton assembly and progression of endocytosis [98,99,100,101]. Apl3 is the large subunit of the clathrin-associated protein complex (AP-2) mediating clathrin recruitment to the endocytic site and connecting cargoes to the clathrin coat [102]. Finally, Lsb3 and Aim21 have been implicated in the regulation of actin assembly during endocytosis [103,104,105,106,107,108]. Thus, our findings suggest that Mck1 may also be involved in the regulation of different stages of clathrin-mediated endocytosis (Figure 7B).

Besides endocytosis and exocytosis, Mck1-dependent phosphorylation was also found in a number of proteins involved in cytokinesis (Table 2). The Gin4 kinase, together with the LKB1/PAR-4-related kinase Elm1, controls septin assembly and stability during bud emergence and enables timely remodeling of the septin hourglass into a double ring during mitosis [109]. Bni5 tethers myosin-II to septins to enhance retrograde actin flow and cytokinesis [110]. Vhs2 maintains the stability of the double septin ring structure until the telophase [111]. Cyk3 is a component of a protein complex involved in the coordination of primary and secondary septum formation during cytokinesis [112,113,114,115]. Lastly, Kre6 may catalyze β-1,6-glucan synthesis to drive cell wall maturation during cell growth and division [116]. Together, the above analysis indicated the potential roles of Mck1 in actin cytoskeleton-mediated polarized growth and cytokinesis.

To reveal how polarized growth may be regulated by Mck1, we conducted phenotypic assays by subjecting early-stationary phase cells to cell wall/membrane perturbation agents. As reported before [72,117], the *mck1Δ* mutants displayed severe growth defects in the presence of SDS (Figure 7C). The *slt2Δ* cells, however, had little growth defects in the presence of the detergent and further *SLT2* removal did not seem to aggravate the plasma membrane defects of the *mck1Δ* mutants on SDS (Figure 7C). In contrast, the *slt2Δ* deletants displayed severe growth defects on Congo Red (CR), Calcofluor White (CFW) or caffeine (Figure 7C), consistent with previous reports [118,119,120,121]. Compared to *slt2Δ*, the *mck1Δ* mutants exhibited moderate sensitivity to CFW or CR and modest growth defects on caffeine (Figure 7C). Removal of *SLT2* significantly reduced the fitness of the *mck1Δ* cells on CR and abolished the resistance of the *mck1Δ* cells to CFW or caffeine. Interestingly, removing *SLT2* did not abolish the resistance of the *mck1Δ* mutants to CR, implying that Mck1 may also negatively regulate an Slt2-independent program to overcome CR stress. Nevertheless, these data strongly support the idea that Mck1 is crucially required for plasma membrane integrity upon membrane stress, and the Slt2-dependent cell wall remodeling program is partially mediated through Mck1 in cell wall-stressed cells.

Polarized exocytosis is the major mechanism by which new membrane components are delivered to the cell surface [122,123]. Cell wall synthesis is primarily achieved through a complex process involving the secretion and depositing of glycoproteins, including cell wall-synthetic enzymes (via exocytosis), in the plasma membrane, whereupon wall polysaccharides (β-glucan and chitin) are made and cross-linked [2]. Furthermore, the coupling of endocytosis with exocytosis enables the targeting and dynamic relocalization of cell wall-synthetic and -remodeling systems [4,82,124]. Together, these studies and our data support the claim that Mck1-mediated phosphorylations of multiple proteins involved in actin cytoskeleton-dependent exocytosis and endocytosis (Figure 7A,B) may act together to promote plasma membrane and cell wall homeostasis. How these phosphorylations are regulated in multiple proteins and how they are integrated to impact on polarized growth remain to be elucidated.

## 3. Discussion

The CWI pathway is the major signaling pathway responsible for cell wall homeostasis in yeast. Here, we have revealed the metabolic and actin cytoskeleton-related roles of the Mck1 kinase to facilitate CWI-dependent cell wall remodeling in glucose-starved and cell wall-stressed cells (Figure 8). In response to glucose depletion, Mck1 is required to activate the stress response regulon and the Snf1-dependent metabolic reprogram (Figure 3 ang 4), promoting UDPG accumulation (Figure 2) to enable isotropic cell wall thickening by the CWI pathway (Figure 1). In plasma membrane- and cell wall-stressed cells, Mck1 is necessary to promote actin cytoskeleton-mediated polarized growth to enable plasma membrane homeostasis and the Slt2-dependent cell wall remodeling program (Figure 7). A slightly thickened cell wall was observed in EXP *mck1Δ* cells (Figure 1C), further supporting the roles of Mck1 in polarized growth in vegetative cells.

Phosphoproteome analysis suggest that Mck1 may promote the Snf1-directed metabolic reprogram indirectly through PKA inhibition (Figure 6C) to derepress the activities of the SNF1 complex and directly by phosphorylating the multiple subunits of the SNF1, SAGA and Ccr4-Not complexes (Table 2 and Figure 8). Although the functional implications of these phosphorylations remain to be established, the significant overlap between the Mck1-activated and the Gcn5-promoted transcriptomes (Figure 6A) support the idea that these Mck1-mediated phosphorylations may be required to augment the activity of the SNF1 complex and transcription activation of the metabolic program (Figure 8). Mck1-mediated PKA inhibition may also act to facilitate the functions of Snf1 in the repression of the glycolysis pathway (Figure 8). In response to glucose depletion, the transcription repressor Rgt1 acts in tandem with two paralogous corepressors, Mth1 and Std1, to repress the transcription of Rgt1-targeted genes [125,126,127], including the hexose transporter genes (HXT) and the hexokinase gene *HXK2* revealed in this study (cluster 2 in Figure 3B and Figure 4C). Transcription repression by Rgt1 requires Snf1-dependent phosphorylation and is relieved by PKA-mediated phosphorylation [128,129,130,131]. Furthermore, Snf1 also phosphorylates the adenylate cyclase Cyr1 to negatively regulate PKA-dependent transcription [132]. Thus, Mck1-mediated PKA inhibition may serve to augment the functions of SNF1 in PKA inhibition itself and also in the repression of the downstream glycolysis pathway (Figure 8). Future work is necessary to reveal the key phosphorylation events in the regulation of PKA and the Snf1/SAGA activities and decide how these phosphorylations are integrated to promote the metabolic reprogram.

In response to plasma membrane perturbation, yeast cells arrest their cell cycle at G_1_ in part due to Mck1-mediated degradation of Cdc6, a component of the pre-replicative complexes (pre-RCs) assembled on DNA to license replication origins in M–G_1_ phase [117,133]. Cdc6 stabilization sensitizes cells to SDS treatment to a lesser degree than the *mck1Δ* deletants [117], suggesting other functions of Mck1 in maintaining plasma membrane integrity. The identification of multiple Mck1-dependent phosphoproteins involved in actin cytoskeleton-mediated polarized growth (Figure 7A,B) and the crucial roles of Mck1 in plasma membrane integrity and cell wall remodeling (Figure 7C) indicate that polarized exocytosis may be positively regulated by the Mck1-mediated phosphorylations (Figure 8). Cell growth is significantly slower on caffeine than on CR or CFW (Figure 7C), possibly due to TORC1 inhibition by caffeine [134,135]. Correspondingly, the *mck1Δ* mutants displayed less severe growth defects on caffeine than on CR/CFW (Figure 7C), suggesting that Mck1-mediated polarized cell growth is crucial to cell wall/plasma membrane repair in fast-growing cells. Mck1 phospohtargets have also been implicated in the cell cycle control in our study (Table 2) and Mck1 has been demonstrated to delay mitotic entry by a number of studies [53,136,137]. These findings further suggest that Mck1 may be involved in coordinating polarized growth with the cell cycle progression. pS/pTP is the loose consensus sequence targeted by Cdk1 [138]. The enrichment of SXXXpS/pTP and pSXXXTP motifs in the Mck1-mediated phosphopeptides (Figure 5D) further supports the idea that Mck1 may be involved in cell cycle control by targeting some of those proteins which have been prephosphorylated by Cdk1, as exemplified by the previously confirmed targets of Mck1, e.g., Cdc6 [133] and Eco1 [139,140].

Besides the actin cytoskeleton, Mck1 has been previously implicated in the regulation of astral microtubule function through phosphorylation of kinesin Kip2, which stabilizes astral microtubules (MTs) and facilitates spindle positioning through transport of MT-associated proteins, such as the yeast CLIP-170 homolog Bik1, dynein and the adenomatous-polyposis-coli-related protein Kar9 to the plus ends of astral MTs [141]. Although Kip2 was not identified as an Mck1 phosphotarget in this study, Bik1 and Kar9 were both shown to be dependent on Mck1 for phosphorylation at multiple sites (Table 2). Recently, Mck1 has been shown to function in parallel with the Kin4 kinase to ensure spindle positioning by counteracting the activation of the mitotic exit network (MEN) [136]. Interestingly, Kin4 was also identified as a potential phosphotarget of Mck1 in our study (Table 2). Together, these findings and our data do support the roles of Mck1 in coordinating multiple cytoskeleton-related processes and the mitotic cell cycle.

The predicted number of substrates for GSK3β was over 500 [142], and more than 100 of them have been reported to be phosphorylated by GSK3 [45,46,143]. Substrates of GSK3β can be broadly divided into three groups: metabolism, cytoskeleton architecture and signaling and transcription [45,46,143,144,145]. Mck1-dependent phosphotargets seem to fall into similar functional categories (Table 2 and Appendix A), suggesting functional conservation between Mck1 and mammalian Gsk-3 kinases. Mammalian Gsk-3 is spatially regulated at cortical sites [79,146]. Recent localization studies indicate that Mck1 is not only located throughout the cytoplasm and the nucleus [147] but also at the cortex of emerging and small buds, the bud neck and the spindle poles [141], corresponding to its pleiotropic roles in cell wall remodeling, polarized cell growth and cell cycle control. Further characterization of these phosphotargets will undoubtedly reveal the underlying mechanisms of Mck1 in the various cellular processes in yeast and provide insights into the roles of the mammalian Gsk-3 kinases in the development of numerous diseases, including Alzheimer’s and Parkinson’s.

## 4. Materials and Methods

### 4.1. Strains and Plasmids

Strains carrying single-gene deletions were obtained directly from the BY4742 (*MAT*α *his3Δ1 leu2Δ0 lys2Δ0 ura3Δ0*) mutant library (Open Biosystems, Huntsville, AL, USA). Strains carrying deletions in multiple genes were generated by combining mutations via either mating and dissection or by PCR-mediated gene replacement using drug resistance or nutritional markers [148]. GFP was fused at the C-terminus of endogenous *FBP1* or *PCK1* by PCR-mediated gene integration using pFA6a-GFP-HISMX6 as a template [149].

### 4.2. Phenotypic Assays

Stress resistance was determined as described previously [25] Yeast cells grown to the early stationary phase were harvested and diluted to OD_600 nm_ of 1. They were serially diluted and then spotted onto YPD plates to determine their resistance to different plasma membrane and cell wall perturbation agents.

### 4.3. Fluorescence Detection and Quantification

Yeast cells bearing reporter plasmids were grown overnight in YP liquid medium containing 2% glucose and then inoculated into YP liquid medium containing 0.6% glucose. Fbp1-GFP or Pck1-GFP levels were assayed in cells by flow cytometry (530/30 488 nm for GFP).

### 4.4. Transmission Electron Microscopy (TEM)

Yeast cells were fixed according to the protocol described by Wright [150]. Briefly, 9.5 mL of cell culture (OD_600 nm_ = 0.5 to 1) was prefixed with 9.5 mL of 2× fixative (0.2 M PIPES, pH 6.8, 0.2 M sorbitol, 2 mM MgCl_2_, 2 mM CaCl_2_, 3% glutaraldehyde) at room temperature for 5 min. Cells were then spun down at 1500× *g* for 5 min and resuspended in 9.5 mL of 1× fixative. After incubating overnight at 4 °C, cells were washed by centrifugation and resuspended in 25 mL of sterile water. The washing was repeated 4 times and cells were resuspended in 0.5 mL of sterile water and fixed with 5 mL of 2% potassium permanganate for 5 min. Cells were harvested again and overlaid with 2% potassium permanganate for 45 min. Fixed cells were washed several times in water until no purple color was evident. Cells were en bloc stained with 1% uranyl acetate at room temperature and then dehydrated in graded absolute acetone. After infiltrating in Spurr’s resin and polymerization, samples were incubated in a 60 °C oven for at least 24 h. Ultrathin sections were cut on a Leica Ultracut UCT microtome (Wetzlar, Germany) and stained with uranyl acetate and lead citrate. The obtained sections were viewed on an H-7650 electron microscope (Hitachi, Tokyo, Japan) at 100 kV. Cell images under electron microscope were taken and cell wall thickness was measured using ImageJ 1.51.

### 4.5. UDP-Glucose, TCA Cycle Metabolite Extraction and LC-MS/MS Analysis

UDP-glucose was extracted from yeast cells (50 OD_600 nm_) following the procedures described by Oka and Jigami [151]. Overnight cultures were diluted into 120 mL of YPD in 500 mL flask with a starting OD_600 nm_ of around 0.2. Cells at 6 h, 12 h and 24 h post-inoculation were harvested and washed in dH_2_O three times and recovered by centrifugation. Briefly, 5 mL of ice-cold 1M formic acid saturated with 1-butanol was added to cell pellets and incubated for 1 h at 4 °C. The supernatant was collected by centrifugation at 4000 g for 10 min and then lyophilized by freeze-drying, and the metabolites were redissolved in 300 μL of H_2_O. Samples were filtered using a syringe filter (0.22 μm, Restek, Bellefonte, PA, USA) by centrifugation at 10,000× *g* for 1 min and then kept at −80 °C until the assay with mass spectrometry. In total, 150 µL of the aqueous extract was lyophilized using a centrifugal evaporator (Labconco, Kansas City, MO, USA) and reconstituted in 300 µL of 70:30 acetonitrile/0.1 M aqueous ammonium carbonate water containing 2 µM [^13^C_10_, ^15^N_5_] adenosine monophosphate as the internal standard. The resulting solution was vortexed then sonicated for 15 min followed by centrifugation at 15,000 rpm to pellet any remaining undissolved material. After centrifugation, 100 µL of the supernatant was transferred into a 300 µL vial and capped ready for analysis using a Quantiva triple stage quadrupole mass spectrometer (Thermo Fisher Scientific, Hemel Hempstead, Herts, UK) coupled with a Vanquish ultra-high performance liquid chromatography system (Thermo Fisher Scientific, Waltham, MA, USA). The strong mobile phase (A) was 100 mM ammonium carbonate, the weak mobile phase was acetonitrile (B) and the LC column used was the BEH amide column (100 × 2.1 mm, 1.7 µm, Waters, Atlas Park, Manchester, UK). The following linear gradient was used: 20% A in acetonitrile for 1.5 min followed by an increase to 60% A over 2.5 min with a further 1 min at 60% A, after which the column was re-equilibrated for 1.9 min. The total run time was 7 min, the flow rate was 0.6 mL/min, and the injection volume was 5 µL. UDP-glucose and the labeled adenosine monophosphate as the internal standard were measured in negative ion mode using unscheduled SRMs (*m*/*z* 565.1→323.0 and *m*/*z* 361.1→144.1, respectively) with a 0.3 s cycle time. The source parameters used were a vaporizer temperature of 440 °C and an ion transfer tube temperature of 362 °C; an ion spray voltage of 3.5 kV; and a sheath gas, auxiliary gas and sweep gas of 54, 17 and 2 arbitrary units, respectively. Samples were acquired, processed and integrated using Xcalibur (Thermo Fisher Scientific) where UDP-glucose peak areas were normalized to those of labeled adenosine monophosphate and presented as relative ratios to that in WT cells.

Extraction of the TCA cycle and other intracellular metabolites in PDS cells and the follow-up data analysis were detailed recently [62].

### 4.6. Analyses of the Transcriptome Data

Total RNA was isolated from exponentially growing and early post-diauxic phase cells using an RNeasy Mini kit (Qiagen, 74104, Hilden, Germany). PolyA-enriched RNA samples were sequenced on the Illumina Hiseq platform by Novogen UK limited (Cambridge, UK). Raw RNA-seq data have been deposited in BioStudies with the accession number E-MTAB-11024. The RNA-seq dataset was analyzed in R4.1.1 as described previously [62].

### 4.7. Total Proteome Isolation and Digestion

Freshly grown overnight cell cultures were inoculated into 400 mL of YPD in 2 L flasks with the starting OD_600 nm_ of 0.1. After grown to mid-exponential (~6 h) and early post-diauxic shift (13 h) phases, cell cultures equivalent to 250 OD_600 nm_ were harvested, and the pellets were rinsed with 10 mL of ice-cold H_2_O, 10 mL of cold 10%TCA (Sigma-Aldrich, T9159, St. Louis, MO, USA), 20 mL of cold PBS (1×), 10 mL of cold Acetone (Sigma-Aldrich) and finally 20 mL of cold PBS (1×) sequentially. Cell pellets were resuspended in residual PBS buffer and transferred into 2 mL screw cap tubes. After being rinsed with 1 mL of cold 0.2 M NaOH, cells were lysed by bead-beating in lysis buffer (8 M urea in 50 M Ammonium Bicarbonate containing PhosSTOP and cOmplete™ inhibitors) as described previously [62]. After removing cell debris by centrifugation (17,000× *g*) for 10 min at 4 °C, total protein concentration in the supernatant was quantified using Pierce^TM^ BCA protein kit (Themofisher).

Freshly prepared DTT stock (0.5 M in 50 mM Ammonium Bicarbonate) was added into 5.5 mg of protein samples to a final concentration of 5 mM and the mixture was incubated for 45 min in the thermomixer (Eppendorf ThermoMixer, 5350, Hamburg, Germany) set at 37 °C and 500 rpm. Following a short spin down (2000 g for 5 min at RT), freshly prepared IAA (Iodoacetamide) stock (0.5 M in 50 mM ammonium Bicarbonate) was added into the protein samples to a final concentration of 10 mM and incubated at room temperature for 30 min in the dark with gentle agitation. DTT stock was then added to the final concentration of 5 mM and the mixture was incubated at room temperature for 30 min with gentle agitation. Protein concentration was quantified again using Pierce^TM^ BCA protein kit (Themofisher). For this, 5 mg of protein samples (8 M urea in lysis buffer) was further diluted to 1.5 M urea with 50 mM ammonium bicarbonate before digestion. Trypsin/Lys-C mix protease (Thermo Fisher, A40009) was added to the protein samples in a ratio of 1:100 (*w*/*w*) to predigest the proteins at 37 °C for 4 h. Further, trypsin (ThermoFisher, 90058) was added to the pre-digested protein samples in a ratio of 1:50 (*w*/*w*) to completely digest the proteins (16 h, 37 °C). The digestion was stopped by adding trifluoroacetic acid (TFA) to the final 0.2% TFA concentration (*v*/*v*), followed by centrifugation at 10,000× *g* for 2 min at room temperature. The supernatant was transferred to fresh tubes and then stored at −80 °C. Peptides from each sample were quantified (ThemoFisher, 23275).

### 4.8. TMT Labeling and LC-MS/MS for Proteome and Phosphoproteome

The peptide samples were desalted using the ultra-microspin silica C18 column (The Nest Group, Ipswich, MA, USA) and dried using a SpeedVac vacuum centrifuge concentrator (Thermo Fisher). The desalted peptides were then labeled with a TMT 11-plex labeling kit according to the manufacturer’s protocol (Themofisher). TMT-labeled samples were combined, acidified and dried. For phosphoproteome analysis, phosphopeptides was enriched using TiO_2_ microspheres after TMT labeling, following the protocol provided by the manufacturer (GL Sciences, Tokyo, Japan). The peptide mixture was de-salted using the ultra-microspin silica C18 column, dried and redissolved in 0.1% formic acid before LC-MS analysis.

All samples were analyzed by using an Orbitrap Fusion Lumos Tribrid mass spectrometer (Thermo Fisher Scientific), equipped with a Dionex ultra-high-pressure liquid chromatography system (RSLCnano 3000, Thermo Fisher). Peptides were injected onto a 75 μm × 2 cm PepMap-C18 pre-column and separated using EASY-Spray columns (C18, 2 µm, 75 µm × 50 cm) with an integrated nano electrospray emitter at a flow rate of 300 µL/min. Peptides were separated with a 180 min segmented gradient as follows starting from 8% to 30% buffer B in 135 min, 30–45% in 20 min and 45–90% in 5 min. The mobile phases were H2O incorporating 0.1% FA (solvent A) and 80% ACN incorporating 0.1% FA (solvent B).

For the total proteome analysis, the emitter spray voltage was set to 2 kV. The RF lens was set at 30%, and the ion transfer tube temperature was set to 275 °C. The Orbitrap Fusion Lumos was operated in positive ion data dependent (DDA) mode with synchronous precursor selection (SPS)-MS3. The data were acquired under the control of Xcalibur software 4.2 using the top speed mode with 3 s per cycle. The full scan was performed in the range of 350–1500 *m*/*z* at a nominal resolution of 120,000 at 200 *m*/*z* and AGC set to 4 × 10^5^ with a maximum injection time of 50 ms, followed by selection of the most intense ions above an intensity threshold of 5 × 10^3^ for MS2. Dynamic exclusion was set to 60 s. Monoisotopic precursor selection was set to peptide. Charge states between 2 and 7 were included for MS2 fragmentation. The isolation width was set to 0.7 *m*/*z* with no offset. Peptide ions were fragmented using collision-induced dissociation (CID) with 35% normalized collision energy (NCE). The MS2 scan was acquired in the ion trap with auto normal range scan and AGC target of 1 × 10^4^. The maximum injection time for MS2 scan was set to 50 ms. For the MS3 scan, SPS was enabled. MS3 was performed in the Orbitrap over 5 notches at a resolution of 50,000 at 200 *m*/*z* and AGC set to 5 × 10^4^ with maximum injection time 105 ms, over a mass range of 100–500 *m*/*z*, with high collision-induced dissociation (HCD) and 65% normalized collision energy.

For phosphoproteome analysis, MS data were acquired in DDA mode with high-resolution MS2 for TMT reporter ion quantification. The mass spectrometer was operated in top speed mode with 3 s per cycle. The full scan was performed in the range of 350–1500 *m*/*z* at nominal resolution of 120,000 at 200 *m*/*z* and AGC set to 4 × 10^5^ with maximal injection time of 50 ms, followed by selection of the most intense ions above an intensity threshold of 5 × 10^4^ for high collision-induced dissociation (HCD)-MS2 fragmentation with 38% NCE. The isolation width was set to 1.2 *m*/*z* with no offset. Dynamic exclusion was set to 60 s. Monoisotopic precursor selection was set to peptide and only charge states between 2 and 7 were considered for MS2 fragmentation. The MS2 scan was performed in the Orbitrap using 50,000 resolutions with auto normal range scan from *m*/*z* 100 to 500 and AGC target of 5 × 10^4^. The maximal injection time for MS2 scan was set to 120 ms.

### 4.9. Proteome and Phosphoproteome Data Analysis

Raw MS data for total proteome and phosphoproteome, deposited to ProteomeXchange with identifier PXD055585 via the PRIDE database, were analyzed together using the MaxQuant computational platform (version 1.6.17.0) with the integrated Andromeda search engine, following standard protocols [152]. Spectra were searched against the UniProtKB Saccharomyces cerevisiae reference proteome database (release 2020_06). Search parameters included carbamidomethylation of cysteine as a fixed modification and oxidation of methionine, N-terminal acetylation and Phospho (STY) (the latter for phosphoproteome only) as variable modifications. Protein identification thresholds were set to a false discovery rate (FDR) of 1% at both peptide and protein levels. The resulting proteome and phosphoproteome data were further analyzed using Perseus (version 2.1.3.0). Contaminants, reverse hits, and proteins identified only by site were excluded from the dataset. For phosphoproteome, those with a phosphorylation probability less than 0.75 were removed. Median normalization was applied for both proteome and phosphoproteome datasets. Missing values were imputed based on the normal distribution of the dataset, and differentially regulated proteins or phosphopeptides were revealed by Limma (Bioconductor version: Release 3.20) [153] in R4.4.2with a permutation-based FDR correction (FDR < 0.05).

## Figures and Tables

**Figure 1 ijms-26-03534-f001:**
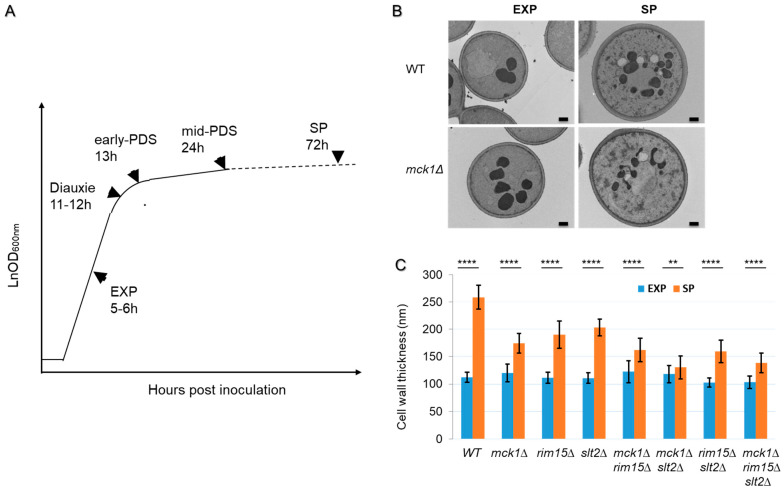
Mck1 and Slt2 act cooperatively to promote cell wall thickening in PDS cells. (**A**): Typical cell growth curve and the sampling timepoints in the study (from the mid-exponential (EXP) to early stationary phase (SP)). (**B**): Representative TEM images of the cell wall of WT and the *mck1Δ* mutant cells. (**C**): Cell wall thickness quantified by TEM imaging analysis. For each strain, cell wall thickness (the electron-transparent layer) at 3 positions of 50 individual cells was determined and averaged. Error bars indicate s.d. among the 50 measurements. Significance of differences between EXP and SP samples was revealed by Student’s *t*-test. ** 0.001 < *p* < 0.01, **** *p* < 0.0001. Bar size: 0.5 µm.

**Figure 2 ijms-26-03534-f002:**
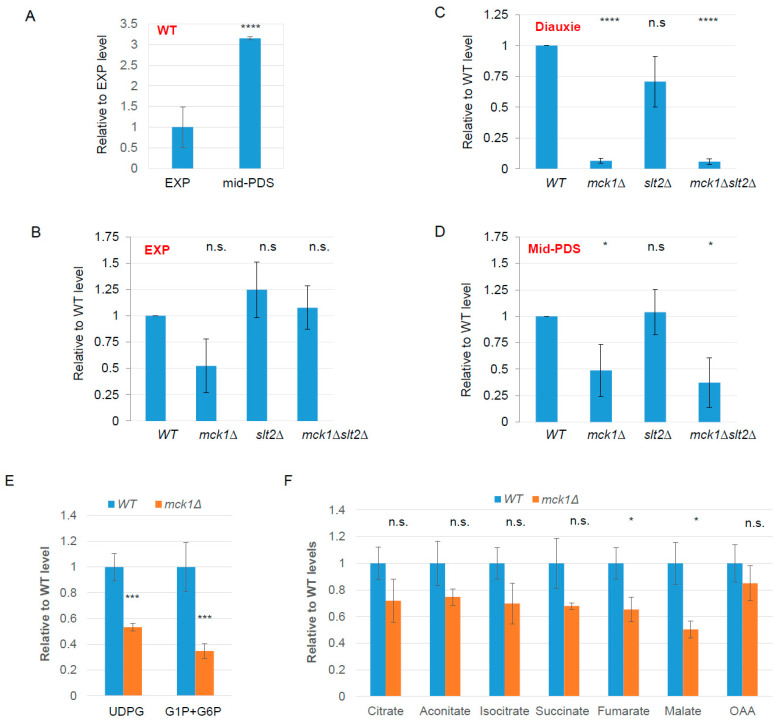
Mck1 is essential to UDPG accumulation in PDS cells. (**A**) Relative levels of UDPG in WT cells. (**B**–**D**) Relative levels of UDPG in the *mck1Δ*/*slt2Δ* mutants grown at EXP (**B**), diauxie (**C**) and mid-PDS (**D**) phases. (**E**,**F**) Relative levels of G1P and G6P (**E**) and TCA cycle intermediates (**F**) in mid-PDS cells. Significance of difference between each of the mutants and WT was revealed by Student’s *t*-test. n.s.: not significant, * 0.01 < *p* < 0.05, *** 0.001 < *p* < 0.0001, **** *p* < 0.0001. OAA: Oxaloacetic acid.

**Figure 3 ijms-26-03534-f003:**
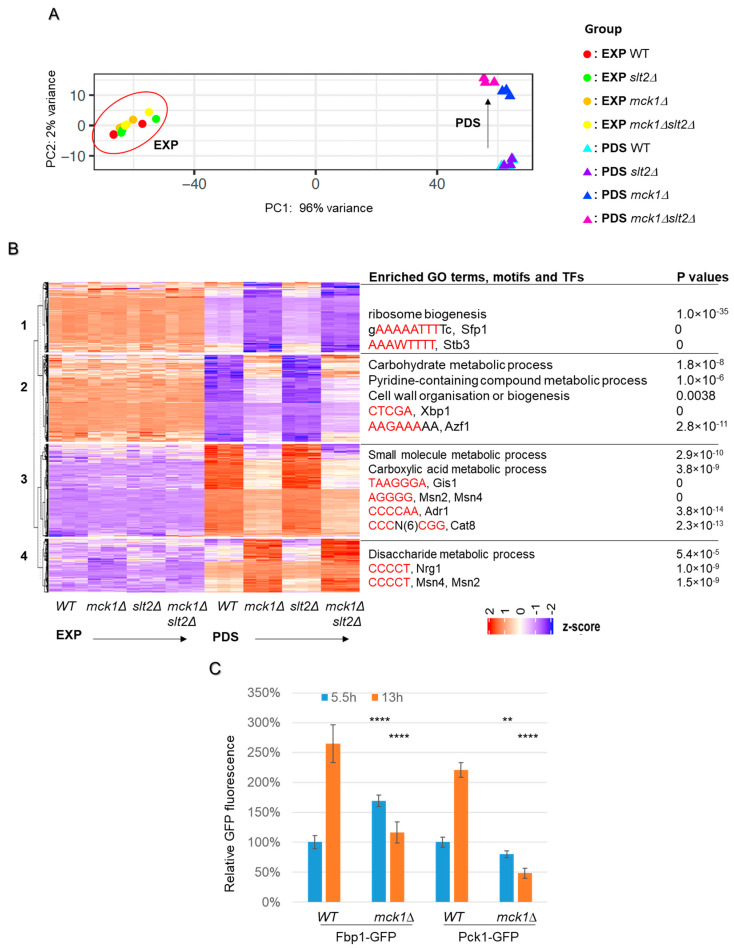
Mck1 regulates a glucose starvation-induced transcription program largely independent of *SLT2*. (**A**) Principal component analysis (PCA) of the transcriptome isolated from cells growing at the mid-exponential (EXP) and early-PDS (PDS) phases. (**B**) Hierarchical clustering and bioinformatics analysis of the differentially expressed (DE) genes regulated by *MCK1* in PDS cells (FDR < 0.05, fold change > 1.5). Enriched motifs revealed by RSAT analysis are highlighted in red in the consensus sequences targeted by the indicated TFs. *p* values indicate the significance of the representation of the GO terms or TFs associated with each gene cluster. (**C**) Relative Fbp1-GFP and Pck1-GFP levels detected in EXP and PDS cells. Significance of difference between the mutants and WT cells was revealed by Student’s *t*-test (**C**). ** 0.001 < *p* < 0.01, **** *p* < 0.0001.

**Figure 4 ijms-26-03534-f004:**
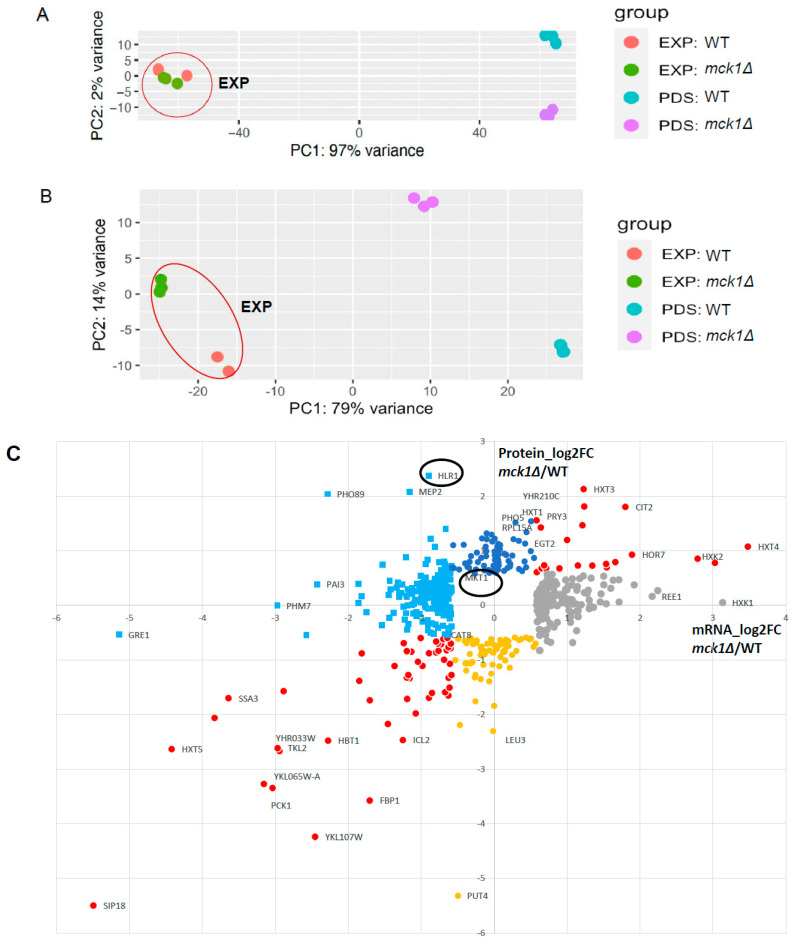
Integrative analysis of the proteome and transcriptome regulated by Mck1. (**A**,**B**) PCA of the transcriptome (**A**) and the proteome (**B**) in all samples. (**C**) Correlation between the transcript and protein regulation levels. Red dots represent genes significantly up-regulated (top right) or down-regulated (bottom left) at both levels in the *mck1Δ* mutants. Among those not significantly regulated at the transcript level, blue and yellow dots denote the significantly up-regulated (blue) and down-regulated (yellow) genes at the protein level. Conversely, gray and green dots represent the transcriptionally up-regulated (gray) and down-regulated (orange) genes without corresponding changes at the protein level.

**Figure 5 ijms-26-03534-f005:**
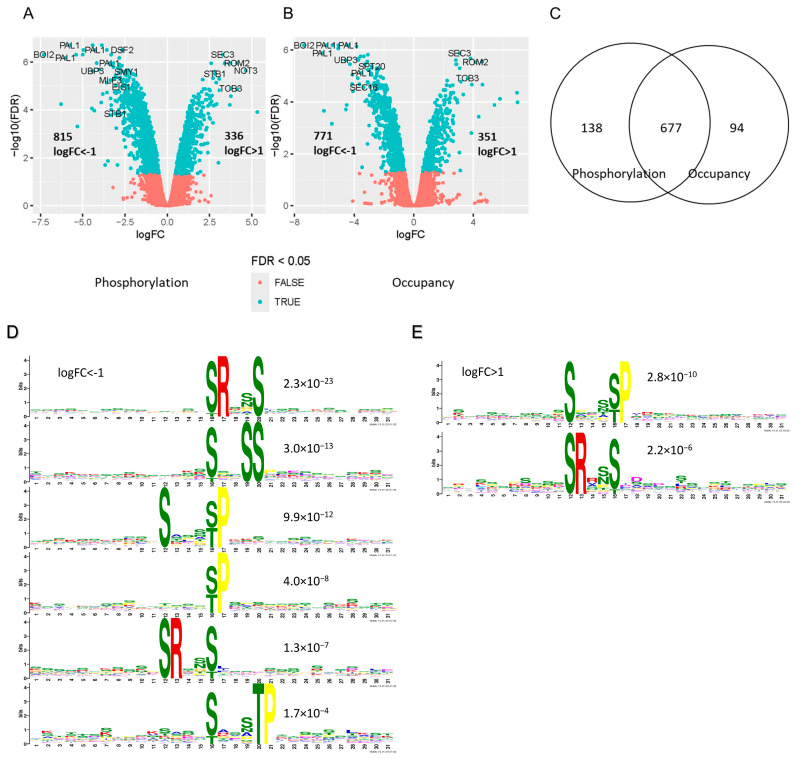
Identifying the Mck1-mediated phosphopeptides. (**A**,**B**) Volcano plots displaying differentially regulated phosphopeptides at the phosphorylation (**A**) and occupancy (**B**) levels (FDR < 0.05); (**C**) Overlap between the phosphopeptides exhibiting reduced phosphorylation and occupancy levels in the *mck1Δ* mutants (logFC < −1). (**D**,**E**) Sequence motifs enriched in the peptides with phosphorylation levels significantly reduced (**D**) or enhanced (**E**) in the *mck1Δ* mutants.

**Figure 6 ijms-26-03534-f006:**
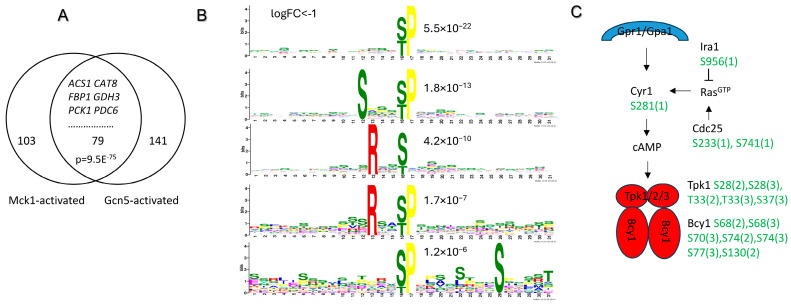
Mck1 may act through SAGA and PKA to facilitate transcription activation of the Snf1-mediated metabolic reprogram. (**A**) Overlap between the Mck1- and Gcn5-activated transcriptome; (**B**) sequence motifs enriched in the phosphopeptides down-regulated in glucose-depleted *mck1Δ* mutants; (**C**) Mck1-mediated phosphorylations in the PKA pathway. The multiplicity of phosphorylation is included in the brackets.

**Figure 7 ijms-26-03534-f007:**
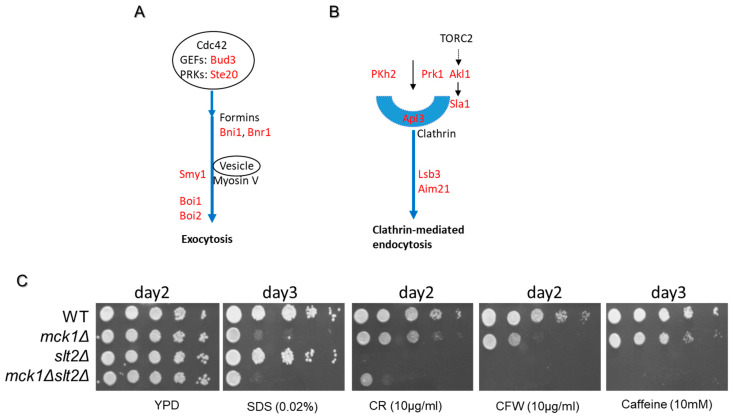
Mck1 is required for polarized growth and cell wall homeostasis in response to plasma membrane and cell wall perturbations. (**A**,**B**): Potential Mck1 phosphotargets involved in polarized exocytosis (**A**) and endocytosis (**B**); (**C**) phenotypic assays conducted on plasma membrane or cell wall disturbing agents. Imaging of cell growth was conducted after incubation for 2–3 days, as indicated on top of each image.

**Figure 8 ijms-26-03534-f008:**
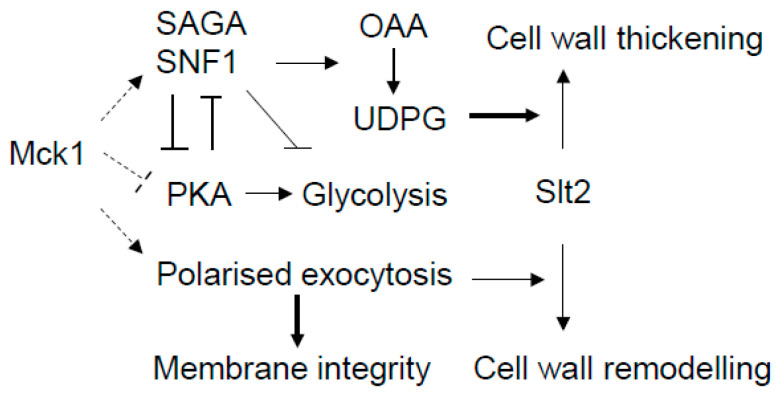
The working model demonstrating the potential targets of Mck1 involved in cell wall remodeling. PKA inhibition and SAGA-mediated transcription activation are proposed as the potential mechanisms of Mck1 in facilitating the Snf1-directed metabolic reprogram (including repression of the glycolysis) in glucose-depleted cells. polarized exocytosis may be promoted by Mck1 to ensure plasma membrane homeostasis and to mediate the Slt2-dependent cell wall remodeling program in cell wall-stressed cells. OAA (Oxaloacetate) to UDPG denotes the metabolic reprogram. Arrow: activation; bar: inhibition; dashed line: phosphorylation–function relationship to be established.

**Table 1 ijms-26-03534-t001:** List of Mck1-activated genes involved in the stress response and metabolism.

Functional Category	Gene Name	log2FC RNA	log2FC Protein	Protein Name
Stress response	*CTA1* *	−0.67	−1.59	Catalase A
	*CTT1*	−1.24	−0.69	Catalase T
	*GPX1*	−0.73	−0.71	Glutathione peroxidase
	*HSP12*	−0.89	−0.88	Heat shock protein
	*SIP18*	−5.50	−5.50	Phospholipid-binding hydrophilin
	*SPG4*	−1.19	−1.72	Stationary phase protein
	*SSA3*	−3.64	−1.70	Heat shock protein
	*SSA4*	−0.66	−0.82	Heat shock protein
Mitochondrial function	*ACO1*	−1.14	−0.85	Aconitate hydratase
	*CYB2* *	−1.71	−1.74	Cytochrome b
	*DLD1* *	−1.05	−1.04	D-lactate dehydrogenase
	*GUT2* *	−1.20	−0.84	Glycerol−3-phosphate dehydrogenase
	*ICL2* *	−1.25	−2.47	2-methylisocitrate lyase
	*IDH1*	−0.59	−0.70	Isocitrate dehydrogenase subunit
	*IDP2* *	−1.20	−1.32	Isocitrate dehydrogenase
	*MBR1*	−0.59	−1.28	Mitochondrial biogenesis regulation protein
	*NDE2*	−1.18	−1.28	External NADH-ubiquinone oxidoreductase
Gluconeogenesis	*FBP1* *	−1.71	−3.58	Fructose−1,6-bisphosphatase
	*PCK1* *	−3.04	−3.35	Phosphoenolpyruvate carboxykinase
	*CAT8*	−0.52	−0.83	Transcription activator
Pentose phosphate pathway	*GND2*	−2.89	−1.57	6-phosphogluconate dehydrogenase
	*TKL2*	−2.97	−2.62	Transketolase 2
Peroxisomal function	*MLS1* *	−0.85	−1.60	Malate synthase
	*YPL113C*	−0.80	−0.66	Glyoxylate reductase
	*FOX2* *	−0.98	−1.11	3-hydroxyacyl-CoA dehydrogenase and enoyl-CoA hydratase
Other metabolic processes	*ACS1* *	−0.89	−1.69	Acetyl-coenzyme A synthetase
	*ADH2* *	−1.37	−1.11	Alcohol dehydrogenase
	*AGX1*	−1.08	−1.98	Alanine-glyoxylate aminotransferase
	*GUT1* *	−0.80	−0.86	Glycerol kinase
	*HXT5*	−4.42	−2.63	Glucose transporter
	*INO1*	−1.16	−1.34	Inositol-3-phosphate synthase
	*NQM1*	−1.82	−0.88	Transaldolase
	*RGI2* *	−1.85	−1.38	Respiratory growth induced protein
	*YKL107W*	−2.46	−4.24	NADH-dependent aldehyde reductase
Other cellular processes	*AMS1*	−0.78	−0.70	Alpha-mannosidase
	*ATG34*	−0.70	−0.62	Autophagy-related protein
	*GAC1*	−0.63	−0.77	Regulatory subunit for Glc7p
	*HBT1*	−2.28	−2.48	Shmoo tip protein
	*MUB1*	−0.60	−0.79	MYND-type zinc finger protein
	*FMP45*	−3.84	−2.06	SUR7 family protein
	*NAT4*	−0.65	−0.82	Histone-specific N-acetyltransferase
	*PNS1*	−0.76	−0.83	Putative choline transporter
	*PRY1*	−0.63	−1.65	Lipid binding protein
	*UIP4*	−0.64	−0.60	Protein required for nuclear envelope integrity
	*XBP1*	−1.46	−2.17	Transcriptional repressor induced by stress or starvation
Unknown	*YBR241C*	−0.61	−1.51	Putative Transporter
	*YDL199C*	−0.61	−1.07	Putative transporter
	*YGR067C* *	−0.62	−1.33	Zinc finger protein
	*YGR201C*	−1.01	−0.60	Putative elongation factor
	*YHR033W*	−2.94	−2.67	Uncharacterized
	*YKL065W-A*	−3.16	−3.27	Uncharacterized
	*YNL195C*	−0.69	−1.00	Uncharacterized

* bound and transcriptionally activated by Cat8 and/or Adr1.

**Table 2 ijms-26-03534-t002:** List of Mck1 phosphoproteins involved in signaling *, Pol II transcription ^&^, cell cycle and polarized growth.

GO Terms	Gene Name	Function	log2FC < −1 (Multiplicity)	log2FC > 1 (Multiplicity)
Signaling *	*LCB5*	Minor sphingoid long-chain base kinase	S160(2); S164(2)	
	*MDS3*	Negative regulator of early meiotic gene expression	S602(2); S603(2); S698(1); S698(2); S702(1)	S606(1)
	*PKH2*	Serine/threonine protein kinase involved in sphingolipid-mediated signaling pathway that controls endocytosis	S988(3); S990(3); S992(3); T997(2); S1001(2); S1003(1); S1003(2); S1005(2)	
	*RCN1*	Noncompetitive calcineurin inhibitor involved in calcium-mediated signaling	S113(2); S117(2); S117(1)	
	*RTC1*	Subunit of SEACAT inhibiting the TORC1 inhibitory role of the Iml1p/SEACIT subcomplex	S1129(2); S1133(2)	S1133(1)
	*SIP1*	One of three alternate beta-subunits of the Snf1p kinase complex	S377(2); S381(2); S385(1)	
	*SIP2*	One of three alternate beta subunits of the Snf1 kinase complex	S66(2); S70(2); S133(2); S133(3); S136(2); S136(3); S137(3)	S137(1)
	*SLN1*	Histidine phosphotransfer kinase	S386(2); S390(2)	
Pol II transcritption ^&^	*ASH1*	Component of the Rpd3L histone deacetylase complex	T87(2); S91(2); S91(3); T104(3); S108(3)	
	*DCP2*	Catalytic subunit of Dcp1p-Dcp2p decapping enzyme complex	S724(2); S725(1); S725(2); S728(2); S729(2)	S729(1)
	*EAF7*	Subunit of nuclear NuA4 histone acetyltransferase complex	S393(1); S393(2);T396(2); S397(2)	T396(1); S397(1)
	*HAA1*	Transcriptional activator involved in adaptation to weak acid stress	S413(2); S417(2)	
	*NOT3*	Component of the CCR4-NOT core complex	S303(2); S304(2); T305(2); S442(2); S442(3); S442(1); S446(2); S446(3); S450(2); S450(3); T454(3)	S307(1)
	*NOT5*	Component of the CCR4-NOT core complex	S271(2); S273(3); S275(2); S275(3); S302(1); S302(2); T306(2)	
	*ROX3*	Subunit of the RNA polymerase II mediator complex	S200(1); S200(3); T204(3)	T204(1)
	*SGF29*	Component of the HAT/Core module of the SAGA, SLIK, and ADA complexes	S83(2); S83(1)	T87(1)
	*SMY2*	ER membrane protein involved in ER-to-Golgi vesicle-mediated transport	S80(2); S80(3); T82(3); S83(2); S83(3); S84(2); S84(3)	
	*SPT20*	Subunit of the SAGA transcriptional regulatory complex	S593(1); S593(2); S595(1); S595(2); T597(2)	
	*STB3*	Transcription activator involved in positive regulation of transcription by glucose	S337(1); S337(2); S341(2)	
	*TAF5*	subunit of SAGA and transcription factor TFIID complex	S411(2); S411(3); S411(1); S414(2); S414(3); S415(2); S415(3)	S414(1); S415(1)
	*WAR1*	Transcription factor; binds to a weak acid response element to induce transcription of PDR12 and FUN34	S124(2); T128(2)	
Polarized growth	*AIM21*	Subunit of a complex associating with actin filaments	S145(2); S149(2)	
	*AKL1*	Ser/Thr protein kinase negatively regulating endocytosis	S403(2); S407(2)	
	*APL3*	Alpha-adaptin	S723(2)	T727(1)
	*BCK1* *	MAPKKK in the PKC1 signaling pathway	S505(2); S509(2)	
	*BNI5*	Linker protein for recruitment of myosin to the bud neck	S270(1); S270(2); S273(2); T274(2)	T274(1)
	*BOI1*	Protein involved in polar growth	S574(2); S574(1); S578(2)	
	*BOI2*	Protein involved in polar growth	T612(2); S615(2); S615(3); S616(3); S617(2); S617(3); S619(3); S620(3); S637(3); S639(2); S639(3); T641(3); S642(2); S642(3); S643(3)	
	*DSF2*	Deletion suppressor of mpt5 mutation; relocalizes from bud neck to cytoplasm upon DNA replication stress	S391(2); S395(2)	
	*KRE6*	Glucosyl hydrolase required for beta-1,6-glucan biosynthesis	S108(2); S108(1); S112(2)	
	*LSB3*	Protein involved in actin cortical patch localization	S377(2); T393(2); S397(2); S397(3)	S381(1)
	*PAL1*	Protein of unknown function thought to be involved in endocytosis	S186(1); S186(2); S186(3); S189(2); S189(3); S190(2); S190(3)	
	*PRK1*	Ser/Thr protein kinase regulating the organization and function of the actin cytoskeleton	S533(1); S533(3); T537(3); S540(3)	T537(2); S540(2)
	*SEC31*	Component of the Sec13p-Sec31p complex of the COPII vesicle coat	S988(2); S988(3); S988(1); S992(3)	S992(1)
	*SKG1*	Transmembrane protein with a role in cell wall polymer composition	T212(2); T212(3); S215(2); S215(3); S216(2); S216(3)	
	*SLA1*	Cytoskeletal protein binding protein	S473(3); S476(3); S477(3)	
	*SMY1*	Kinesin-like myosin passenger-protein	S566(2); S566(1); S570(2)	S570(1)
	*TAO3*	Component of the RAM signaling network involved in regulation of Ace2p activity and cellular morphogenesis	S318(2)	S322(1)
Polarized growth and cell cycle	*BIK1*	Microtubule-associated protein	T85(1); T85(2); T85(3); S86(1); S86(2); S86(3); T89(2); T89(3); T90(3)	
	*BNI1*	Formin; polarisome component	S75(1); S75(2); S79(2); S257(2); S1334(1); S1334(3); T1337(3); S1338(1); S1338(2); S1338(3)	
	*BNR1*	Formin nucleating the formation of linear actin filaments	S604(2); S604(3); S608(3)	
	*BUD3*	Guanine nucleotide exchange factor (GEF) for Cdc42p	T1026(3); S1029(3); S1030(3)	
	*CDC55* *	Regulatory subunit B of protein phosphatase 2A	S145(1); S145(2); S149(2)	
	*CYK3*	SH3-domain protein located in the bud neck and cytokinetic actin ring	S118(2); S122(2)	S122(1)
	*GIN4* *	Protein kinase involved in bud growth and assembly of the septin ring	S947(3); S949(3)	S950(1); S951(1)
	*HSL1* *	Ser/thr protein kinase involved in the G2/M transition	S1325(2); S1328(2); S1329(2)	
	*KAR9* *	Spindle pole protein	T586(2); T590(2)	
	*KIN4* *	Serine/threonine protein kinase inhibiting the mitotic exit network (MEN) when the spindle position checkpoint (SPOC) is activated	S384(2); S384(3); S388(2); S388(3)	
	*STE20* *	MAP kinase kinase kinase kinase involved in pheromone signaling, bud site selection, regulation of mitotic exit and others	S524(1); S524(2); T528(2)	
Cell cycle	*ACE2 * ^&^	Transcription factor required for septum destruction after cytokinesis	S249(3); S253(3)	S253(1)
	*BCK2*	Serine/threonine-rich protein involved in PKC1 signaling pathway	S575(2); T579(2)	
	*CDC4*	F-box protein required for both the G1/S and G2/M phase transitions	S71(2); S71(3); S74(2); S74(3); T75(2); T75(3)	S74(1); T75(1)
	*IGO1*	Protein required for initiation of the G0 program	S7(1); S11(2)	
	*LTE1*	Protein similar to GDP/GTP exchange factors	S850(2); S854(2)	S854(1)
	*MCM3*	Protein involved in DNA replication	S777(2); S777(1); S779(2); S779(3)	S779(1); S781(1)
	*RFM1 * ^&^	Component of the Sum1p-Rfm1p-Hst1p complex	S211(2); S215(2)	
	*SLD2*	DNA-binding subunit of the DNA replication preinitiation complex	S124(2); S128(2)	S128(1)
	*VHS2*	Regulator of septin dynamics	S84(2)	S88(1)

## Data Availability

Data is contained within the article and Appendix A or publicly available as stated in Section 4.

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
