# Peer review of "The Yeast Gsk-3 Kinase Mck1 Is Necessary for Cell Wall Remodeling in Glucose-Starved and Cell Wall-Stressed Cells"

_ijms, 2025, doi:10.3390/ijms26083534_

Round 1
Reviewer 1 Report
Comments and Suggestions for Authors
The paper provides a detailed description of the role of the yeast Gsk-3 kinase Mck1 in metabolic reprogramming in PDS cells by inhibiting PKA and activating SAGA-dependent transcription in a multi-omics approach. The use of TMT labelling extended the data to the identification of specific phosphopeptides, complementing the transcriptional data. Furthermore, the identification of Mck1-regulated phosphoproteins involved in signalling, transcription, cell cycle and polarized growth will be informative for the yeast research community. The Mck1 protein may be considered to have an important role in cellular mechanisms sustaining membrane integrity and cell wall remodelling involving indirect activation of Slt2.
Overall the ms is well written and contains a wealth of data of possible relevance to other Gsk-3 kinases and the role of Mck1 in yeast metabolism.
Points to be addressed:
The labelling of supplementary File 4 is missing. Labelling of this figure requires more explanation.
Considering the amount of data presented I think the ms would be improved by the inclusion of a paragraph addressing salient points of the investigation. This would clarify the significance and impact of the data.
Author Response
Dear Reviewer,
Thank you very much for your positive comments and suggestions on our manuscript. Please see below our point-by-point responses.
Open Review 1
Points to be addressed:
1. The labelling of supplementary File 4 is missing. Labelling of this figure requires more explanation.
We have now added the labels and explanations to supplemental File 4.
2. Considering the amount of data presented I think the ms would be improved by the inclusion of a paragraph addressing salient points of the investigation. This would clarify the significance and impact of the data.
The significance of the study is now addressed in the last paragraph of the Discussion (lines 595-600).
Reviewer 2 Report
Comments and Suggestions for Authors
Cell wall is one of the important structures of Saccharomyces cerevisiae, which is closely related to the performance of yeast cells. This study presents a compelling multi-omics investigation into the role of the Mck1 kinase in cell wall integrity (CWI) and metabolic reprogramming, offering novel insights into its coordination with the Slt2 MAP kinase pathway. The findings are significant for understanding fungal stress responses and polarized growth. The research content of this article is substantial, especially the construction of a working model of Mck1 kinase and potential targets in the process of cell wall remodeling. This article “The yeast Gsk-3 kinase Mck1 is necessary for cell wall remodeling in glucose-starved and cell wall-stressed cells” can be accepted, but the authors need to answer some questions and make minor revisions.
1. In lines 102-128, the cell wall of Mck1 knockout strain was detected by TEM, and changes in cell wall thickness were found. In addition, the strains analyzed by multi-omics were almost Mck1 gene knockout mutant strains. Why has no attempt been made to express Mck1, analyze cell wall changes, or perform omics studies? Is the effect not significant?
2. In Table 1 (line 286), Mck1 was identified to be associated with the activation of HSP12, SSA3, and SSA4 genes of the heat shock protein family. According to my knowledge, the heat shock transcription factor hsf1 can also regulate the expression of Hsps. So, is there a correlation between Mck1 and Hsf1?
3. In view of the importance of mck1 to the cell wall of S. cerevisiae, can it be applied to the biosynthesis of related product chassis, such as terpenes and aromatic compounds? Do the authors have any future plans to delve further in this direction?
4. This paper focuses on the classic microbe S. cerevisiae. In terms of synthetic biology, there are Yarrowia lipolytica and Pichia pastoris in the yeast chassis. Does mck1 have a protein with similar function in this non-S. cerevisiae? Does it have the same cell wall strengthening function? This could be a huge help in the manufacture of cell factories.
5. The author has 7 signature units, but only 6 are listed in the article.
The authors are advised to thoroughly proofread the throughout manuscript and correct all English mistakes.
Author Response
Dear Reviewer,
We greatly appreciate your positive comments and suggestions on our manuscript. We have now proofread the manuscript thoroughly and corrected all the grammar mistakes and typos. Please see below our point-by-point responses to your questions.
Open Review2
…………….This article “The yeast Gsk-3 kinase Mck1 is necessary for cell wall remodeling in glucose-starved and cell wall-stressed cells” can be accepted, but the authors need to answer some questions and make minor revisions.
1. In lines 102-128, the cell wall of Mck1 knockout strain was detected by TEM, and changes in cell wall thickness were found. In addition, the strains analyzed by multi-omics were almost Mck1 gene knockout mutant strains. Why has no attempt been made to express Mck1, analyze cell wall changes, or perform omics studies? Is the effect not significant?
In all the analyses (TEM and multi-omics), the mck1 knockout mutants were compared to the wildtype strains (in which MCK1 is intact and expressed) at all the growth phases.
2. In Table 1 (line 286), Mck1 was identified to be associated with the activation of HSP12, SSA3, and SSA4 genes of the heat shock protein family. According to my knowledge, the heat shock transcription factor hsf1 can also regulate the expression of Hsps. So, is there a correlation between Mck1 and Hsf1?
In glucose-depleted cells, the stress response transcription factors Msn2/Msn4 and the post-diauxic shift factor Gis1 are dominantly responsible for the activation of HSP12, SSA3 and SSA4. Hsf1 is involved in the activation of some of the heat shock genes whose promoters bear the heat-shock elements (HSEs), such as SSA3. We have previously confirmed the relationship between Hsf1 and Msn2/4/Gis1 in the activation of SSA3 (please see the supporting file1 in PMID: 27923067). Mck1 has been shown to promote nuclear localisation of Msn2 (PMID: 31464369). Whether Mck1 also positively regulate the transcription activity of Hsf1 remains to be determined.
3. In view of the importance of mck1 to the cell wall of S. cerevisiae, can it be applied to the biosynthesis of related product chassis, such as terpenes and aromatic compounds? Do the authors have any future plans to delve further in this direction?
This is a very good question. We are exploring the possibility of using the mck1 mutants to limit cell wall synthesis, thus more efficient production and the recovery of the acetyl-CoA based products, such as terpenes.
4. This paper focuses on the classic microbe S. cerevisiae. In terms of synthetic biology, there are Yarrowia lipolytica and Pichia pastoris in the yeast chassis. Does mck1 have a protein with similar function in this non-S. cerevisiae? Does it have the same cell wall strengthening function? This could be a huge help in the manufacture of cell factories.
This is also a very good question. We know that there are MCK1 orthologues in Yarrowia and Pichia species, but not sure that they have similar functions as Mck1 in S. cerevisiae.
5. The author has 7 signature units, but only 6 are listed in the article.
The 7th one has been added to the list of authors.